# Synergistic Effect of Orange Oil Adjuvant on Acetamiprid in the Control of *Edentatipsylla shanghaiensis*

Guangchun Xu [1,2,3,*], Dongdong Yan [1], Wensheng Fang [1], Dejin Xu [3], Lu Xu [3], Qiuxia Wang [1] and Aocheng Cao [1,*]

1. Institute of Plant Protection, Chinese Academy of Agricultural Sciences, Beijing 100193, China; yandongdong@caas.cn (D.Y.); fangwensheng@caas.cn (W.F.); wqxcasy@163.com (Q.W.)
2. College of Science, China Agricultural University, Beijing 100193, China
3. Institute of Plant Protection, Jiangsu Academy of Agricultural Sciences, Nanjing 210014, China; jaasxdj@jaas.ac.cn (D.X.); xulupesticide@163.com (L.X.)
* Correspondence: xgc551@163.com (G.X.); caoac@vip.sina.com (A.C.)

**Abstract:** We explore the effects of orange oil adjuvant (a kind of spray adjuvant) on the physico-chemical properties of acetamiprid (pesticide) when foliage-applied to the surface of pittosporum tobira leaves. The leaf surface was characterized by the OCG (Van Oss–Chaudhury–Good) method, and the relationship between the wetting behavior of various pesticide droplets, including the change in surface free energy (SFE), adhesion force, and adhesion work, is explored to offer insight into the control of the pittosporum tobira psyllid, *Edentatipsylla shanghaiensis* Li *et* Chen. Results showed that SFE values for the adaxial and adaxial leaf surfaces were 40.13 mJ/m$^2$ and 37.06 mJ/m$^2$, respectively, while acetamiprid liquids had SFE values of 67.43 mJ/m$^2$ and 63.26 mJ/m$^2$. SFE values of the acetamiprid liquids are greater than that of the leaf surface, and the droplets on the leaves with a smaller adhesion force and lager adhesion work exhibited moderate-to-poor wettability estimated by contact angles. When the concentration of the orange oil adjuvant was between 0.10% and 1.00% above CMC (critical micellar concentration, 0.09%), the SFE values of the acetamiprid liquids were less than that of the leaf surface. The adhesion tension was greatly increased, and the adhesion work decreased by 14.46–28.13%. Meanwhile, droplets on the leaves exhibited good wettability. Field experiments showed that the synergistic effect of acetamiprid against *E. shanghaiensis* was significantly improved after spraying with orange oil adjuvant at the concentrations 0.10% and 1.0% above CMC. This study demonstrated the use of an orange oil adjuvant with a concentration above CMC to improve the synergistic effect of the insecticide directly through improved leaf wetting, which can provide reference for reducing pesticide dosage and increasing efficiency during the chemical control of pests.

**Keywords:** adjuvant; surface tension; surface free energy; pittosporum tobira; *Edentatipsylla shanghaiensis* Li *et* Chen





## 1. Introduction

*Edentatipsylla shanghaiensis* Li *et* Chen, also known as pittosporum tobira psyllids, belongs to the Edentate Psyllids genus in the family of Hemipteran Psyllids. In recent years, it has been commonly found on the pittosporum tobira along streets in many regions of China and is characterized by its large quantity and long infestation period, which has seriously affected the landscape effect and the growth of pittosporum tobira plants. *E. shanghaiensis* has an occurrence cycle of five or six generations per year, with multiple generations overlapping each other. Infestations are usually most serious in spring and autumn, according to our survey carried out within the last two years in Jiangsu province, and the damage is mainly caused by adults and nymphs through piercing and sucking young buds and new leaves of pittosporum tobira twigs, resulting in damaged leaves that curl and wrinkle toward the adaxial surface, thus seriously affecting the expansion of the leaf blade. Similar to most psyllids, in addition to piercing and sucking, the excrement

of the *E. shanghaiensis* is often attached to the end of its abdomen, forming a long white waxy filament (as shown in Figure 1, taken on 4 May 2022 from Nanjing, Jiangsu province, China, 118°55′ E, 32°2′ N). This waxy substance is highly sticky and covers the leaf surface of the pittosporum tobira, blocking the stomata of the leaf blades and severely affecting the photosynthesis and respiration of leaves. Moreover, it can also induce the occurrence of sooty blotch [1–3]. Currently, the control of psyllid pests still relies on chemical methods, but there are relatively few reports on the chemical control of *E. shanghaiensis* [4]. The spraying of chemical pesticides diluted with water on the target pests is an essential application method in the process of chemical control [5,6]. During the spraying process, spray adjuvants are often added to reduce the surface tension of the spray liquid, promote the wetting, and promote the spread of the chemical across the surface of the target plants, which thereby increases the deposition of the spray liquid and increases the utilization of the pesticides [7–9].

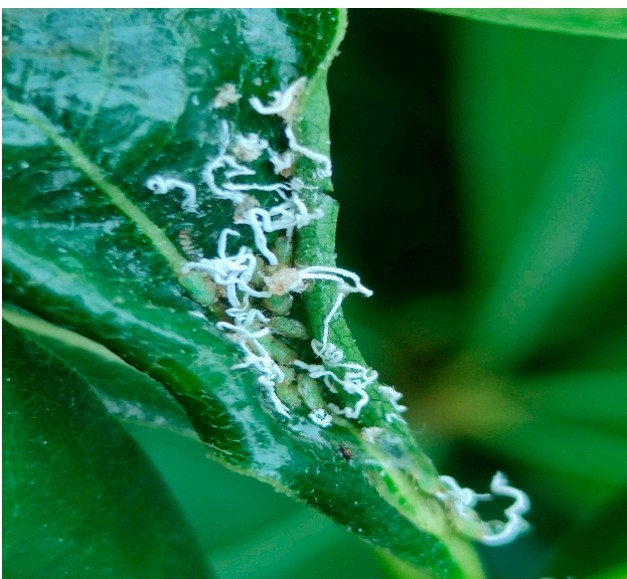

**Figure 1.** Infestation of *Edentatipsylla shanghaiensis* Li *et* Chen on pittosporum tobira leaf.

In recent years, with China's new requirements for food security and for an effective supply of high-quality agricultural products, increasing attention has been paid to the demands for a reduction in the amount and an increase in the synergistic effect of pesticides [10,11]. According to a calculation from the Ministry of Agriculture and Rural Affairs, the utilization of pesticides in China's three major food crops (rice, wheat, and corn) showed an increasing trend, which has reflected the improvement in levels of pesticide use on food crops in China. Compared with food crops, there are relatively few studies on pesticide utilization on other crops. In the practical application scenarios of target spraying control, there are many factors that affect the utilization of pesticides. In addition to the pesticide application apparatus, pesticide application liquid amount, meteorological conditions, and other factors, the target's biological characteristics and the physical and chemical properties of the pesticide liquid are also significant [12–16]. Before flowering, air-assisted sprayers are adopted for pest control in orchards, and the utilization of the pesticides is up to 38% [17]. Due to the high hydrophobicity of the rice leaf and the higher leaf angle, the utilization of pesticides under the conventional large-capacity application is low, but adding adjuvants to the spray liquid may effectively enhance the utilization of pesticides. Moreover, spray adjuvants can also adjust the physical and chemical properties of the pesticide liquid, make the spray droplets better adhere to the target, and wet the spread cloth so as to increase the contact probability between the droplets of the pesticide liquid on the target plant and the target pests, which achieves the objective of improving biological effects [18]. Therefore, in the practical application scenario of pest control, it

is necessary to add spray adjuvants to adjust the physical and chemical properties of the pesticide liquid so as to match the target's characteristics.

As a kind of vegetable oil adjuvant, orange oil adjuvant features good biodegradability and a wide range of applications, whereas conventional nonionic adjuvant and mineral oil adjuvant have no synergistic effect when the humidity is less than 65% and the temperature is more than 28 °C [19]. Although the organic silicone adjuvants commonly used in production have good synergistic effects, they have a narrow pH range (pH 5–8), and they degrade rapidly and lose their effect in strong acid and alkali solutions [20,21]. Orange oils are natural plant products from orange peels with low non-target toxicity and a wide spectrum of activity [22]. Recently, orange oils have been used as spray adjuvants in the agricultural field because of their safety and excellent surface activity [23,24]. The orange oil adjuvant can be used with insecticides, fungicides, and herbicides against pests [25,26]. Laboratory studies showed antifungal activity of the orange oils against five tested pathogens in the range of 15.6–100% [27]. Similarly, *Botrytis cinerea* incidences of 2.9–17.1% and 10.0–30.8% were significantly reduced with the addition of orange oil adjuvants [28]. Adjuvants can alter the contact angle between the droplets and the epicuticular wax layer for a better droplet contact according to target leaf characteristics, but limited studies have reported the use of higher or lower rates of the adjuvant on different leaf surfaces for wetting purposes, as well as the way in which to combine it with insecticides for special pest control. The relationship between the physicochemical properties of pesticide liquid and leaf surface characteristics, as well as the wettability and field efficacy were determined to elucidate the internal mechanism of its synergistic effect. Together with the synergistic interactions between orange oil adjuvant and widely used insecticide, the results could provide insights on how suitable amounts of adjuvant used in the chemical control of *E. shanghaiensis* can reduce pesticide dosage and increase efficiency.

## 2. Materials and Methods

### 2.1. Experiment Materials

The variety of pittosporum tobira used is called "Aisheng", and the reagents used in this experiment include deionized water, 99% glycerol (provided by Shanghai Aladdin Bio-Chem Technology Co., Ltd., Shanghai, China), 98% diiodomethane (provided by Shanghai Aladdin Bio-Chem Technology Co., Ltd., Shanghai, China), 50% acetamiprid water dispersible granules (produced by Sichuan Runer Technology Co., Ltd., Chengdu, China), and orange oil adjuvant (provided by American Oro Agri Agricultural Chemicals Company, Fresno, CA, USA).

### 2.2. Instruments and Software

The following were used for this study: contact angle meter JC2000C1B (produced by Shanghai Powereach Digital Technology Equipment Co., Ltd., Shanghai, China); 0–50 μL microsyringe MS50 (produced by Shanghai Gaoge Industry and Trade Co., Ltd., Shanghai, China); surface tension meter DCAT11EC (produced by Germany Dataphysics Instruments Company, Filderstadt, Germany); Seesa Knapsack electric sprayer SX-MD18DA with a standard sprinkler with 0.3 MPa spray pressure and 0.7 L/min flow rate (provided by Seesa Holdings Co., Ltd., Gauteng, South Africa); and a Hyperpure water system (established by Plant Protection Institute of Jiangsu Academy of Agricultural Sciences, Nanjing, China).

### 2.3. Experiment Method

#### 2.3.1. Determination of Contact Angle on Leaf Surface of Pittosporum Tobira

The following steps should be taken to perform this experiment. Fix the leaf of pittosporum tobira on the object stage of the contact angle meter in a natural state. Use a microsyringe to drop 2 μL of liquid on the leaf, and then capture the droplets on the leaf surface every 10 s with a CCD camera attached to the contact angle meter. Input them into the computer, and calculate the static contact angle of the droplets on the leaf surface of pittosporum tobira by applying the fitting analysis method (about 40 s). Upon

determination, the temperature should be $20 \pm 3\,^{\circ}\mathrm{C}$ and the relative humidity should be $65 \pm 5\%$ [29,30].

### 2.3.2. Determination of Surface Free Energy for Leaf Surface of Pittosporum Tobira

Determine the static contact angle of 3 kinds of detection solutions (deionized water, glycerol, and diiodomethane) on the leaf surface of pittosporum tobira when the temperature is $20 \pm 3\,^{\circ}\mathrm{C}$ and the relative humidity is $65 \pm 5\%$. Then, calculate the surface free energy of the leaf surface according to the OCG method [29–31]. As Van Oss et al. suggested, the free energy of a solid surface can be expressed as the sum of the Lifshitz–van der Waals component $\gamma^{LW}$ (representing the non-polar interaction of surface free energy) and the acid–base interaction component $\gamma^{AB}$ (representing the polar interaction of surface free energy), where $\gamma^{AB}$, in turn, contains a Lewis acid component $\gamma^{+}$ and a Lewis base component $\gamma^{-}$. Therefore, the surface energy of solids or liquids can be calculated using Equations (1) and (2):

$$\gamma_S = \gamma_S^{LW} + \gamma_S^{AB} = \gamma_S^{LW} + 2\sqrt{\gamma_S^{+}\gamma_S^{-}} \tag{1}$$

$$\gamma_L = \gamma_L^{LW} + \gamma_L^{AB} = \gamma_L^{LW} + 2\sqrt{\gamma_L^{+}\gamma_L^{-}} \tag{2}$$

By combining the above Equations (1) and (2), a new relationship between the interface tension and the solid and liquid can be obtained, as shown in Equation (3).

$$\gamma_{SL} = \left(\sqrt{\gamma_S^{LW}} - \sqrt{\gamma_L^{LW}}\right)^2 + 2\left(\sqrt{\gamma_S^{+}\gamma_S^{-}} + \sqrt{\gamma_L^{+}\gamma_L^{-}} - \sqrt{\gamma_S^{+}\gamma_L^{-}} - \sqrt{\gamma_S^{-}\gamma_L^{+}}\right) \tag{3}$$

By combining with the Young equation, we obtain:

$$\left(\gamma_L^{LW} + 2\sqrt{\gamma_S^{+}\gamma_S^{-}}\right)(1 + \cos\theta) = 2\left(\sqrt{\gamma_S^{LW}\gamma_L^{LW}} + \sqrt{\gamma_S^{+}\gamma_L^{-}} + \sqrt{\gamma_S^{-}\gamma_L^{+}}\right) \tag{4}$$

### 2.3.3. Determination of Physical and Chemical Properties of Liquids

Use the software program SCAT31 of the surface tension meter to determine the surface tension $\gamma$ of the prepared and fully stirred liquid, and calculate the adhesion tension β and work of adhesion $W_a$ according to the static contact angle θ and the wetting Equations (5) and (6). Simultaneously, the critical micelle concentration (CMC) of the adjuvants was measured by the software module CMC (SCAT33) of the surface tension meter.

$$B = \gamma \times \cos\theta \tag{5}$$

$$W_a = \gamma \times (\cos\theta + 1) \tag{6}$$

### 2.3.4. Wetting Behavior of Single Droplet on the Leaf of Pittosporum Tobira

Fix the fresh leaf of pittosporum tobira on the object stage with a double-faced adhesive tape. Use a microsyringe to drop 2 μL of droplets on the leaf surface of the pittosporum tobira, and then capture the droplets on the pittosporum tobira leaf surface 40 s after contact using a CCD camera attached to the contact angle meter. Input them into the computer, apply the fitting analysis method to calculate the static contact angle of the droplets on the leaf surface of pittosporum tobira, and analyze its wetting behavior.

### 2.3.5. Field Efficacy Trials

The experiment was carried out in the planting fields of pittosporum tobira in Erling Town, Danyang Town, and Zhenjiang City in Jiangsu Province ($119^{\circ}35'$–$119^{\circ}36'$ E, $31^{\circ}51'$ N). Six treatments were involved, with 600 kg of pesticide liquid per hectare, in the community area of 20 m$^2$, and they were repeated three times and arranged randomly in blocks. To perform this experiment, treat with 120 g/hm$^2$ and 225 g/hm$^2$ of 50% acetamiprid water

dispersible granules. The 120 g/hm$^2$ spray liquid containing 50% acetamiprid water dispersible granules was foliage-applied with orange oil adjuvant at the concentration of 0.01%, 0.10%, and 1.00%. Prepare the water blank control. At the time of our experiment, the pittosporum tobira was in the summer shoot growth stage. The experiment site was surrounded by forests and paddy fields. The weather conditions were cloudy. The temperature was between 25 and 30 °C, and the wind came from southeast at a speed of 0–0.8 m/s. The relative humidity was 66%.

### 2.3.6. Data Analysis

For data analysis, perform a population count before the pesticide application, and survey the remaining pests on the 5th, 10th, and 15th day after the pesticide application. Survey 4 plants of pittosporum tobira in each community. Mark them with tags on damaged young shoots at the eastern, western, southern, northern, and central positions (recording 6 top leaves), and calculate the number of live pests. After the final survey, remove the tagged shoots and retrieve them to calculate the number of live pests. Calculate the reduction rate of the pest population by using the following formula:

$$\text{(Population count before pesticide application} - \text{number of live pest after pesticide application)/population count before pesticide application} \times 100\% \tag{7}$$

Calculate the control rate by using the following formula:

$$\text{(Reduction rate of pest population at treatment area} - \text{reduction rate of pest population at control area)/(100} - \text{reduction rate of pest population at control area)} \times 100\% \tag{8}$$

Then, conduct multiple comparisons of all data after processing with Excel by using Duncan's new multiple range method to analyze the significance of difference.

## 3. Results

### 3.1. Surface Free Energy for Leaf Surface of Pittosporum Tobira

We selected three kinds of liquids, including polar deionized water, polar glycerol, and nonpolar diiodomethane, with surface tensions of 72.80 mN/m, 63.70 mN/m, and 50.80 mN/m, respectively [30]. The value of the liquid surface tension at room temperature and under atmospheric pressure was consistent with that of the surface free energy, so the surface energy was 72.80 mJ/m$^2$, 63.70 mJ/m$^2$, and 50.80 mJ/m$^2$, respectively. The corresponding Lifshitz–van der Waals components $\gamma^{LW}$ were 21.80 mJ/m$^2$, 33.60 mJ/m$^2$, and 50.80 mJ/m$^2$, respectively. The electron acceptor components $\gamma^+$ were 25.50 mJ/m$^2$, 8.41 mJ/m$^2$, and 0.56 mJ/m$^2$, respectively, and the electron donor components $\gamma^-$ were 25.50 mJ/m$^2$, 31.16 mJ/m$^2$, and 0.00 mJ/m$^2$ [29,30], respectively. We dropped single droplets of different solutions on the horizontally fixed leaves of pittosporum tobira, and the static contact angles obtained are shown in Table 1. It can be seen from the table that the static contact angles of two kinds of liquids (water and glycerol) on the leaves of pittosporum tobira were more than 80° with no significant difference, but those of diiodomethane were all less than 60° with a significantly lower difference than water and glycerol. By calculating according to the OCG method, the SFE values of the adaxial and abaxial leaf surfaces were 40.13 mJ/m$^2$ and 37.06 mJ/m$^2$, respectively. The corresponding Lifshitz–van der Waals components $\gamma^{LW}$ were all higher than the acid–base components $\gamma^{AB}$, indicating that the nonpolar interactions dominate over the polar ones. Moreover, the SFE values of the leaves were all less than 100 mJ/m$^2$, so the leaf surfaces of pittosporum tobira were of low energy.

**Table 1.** Surface free energy of *Pittosporum Tobira* leaf surface.

| Leaf of Pittosporum Tobira | Static Contact Angle (°) | | | Surface Free Energy $\gamma_s$ (mJ/m²) | $\gamma^{LW}$ (mJ/m²) | $\gamma^{AB}$ (mJ/m²) |
|---|---|---|---|---|---|---|
| | Water (W) | Glycerol (G) | Diiodomethane (DM) | | | |
| Adaxial Leaf | 83.41 ± 8.14 a | 82.03 ± 1.84 a | 47.22 ± 7.90 b | 40.13 | 30.93 | 9.20 |
| Abaxial Leaf | 96.96 ± 7.18 a | 92.31 ± 6.10 a | 52.55 ± 7.89 b | 37.06 | 29.71 | 7.35 |

Note: The data in the table are mean ± standard deviation ($n = 5$). Different letters in the same row indicate significant difference at the level of $p < 0.05$ by Duncan's new multiple range test.

### 3.2. Influence of Orange Oil Adjuvant on Properties of Spray Liquids

At room temperature and atmospheric pressure, the value of the liquid surface tension (characterizing surface phenomena from the perspective of force) and the liquid surface free energy (characterizing surface phenomena from the perspective of energy) were consistent, but with different dimensions, and their units were different as well. The CMC of the orange oil adjuvant determined by the surface tension meter is 0.09% and its corresponding surface tension is 29.33 mN/m. The surface tension of the 120 g/hm² acetamiprid formulation was 67.43 mN/m (with a surface free energy of 67.43 mJ/m²); foliage-applied with orange oil adjuvant significantly reduced the value to 29.85 mN/m (with a surface free energy of 29.85 mJ/m²). In addition, the surface tensions of the liquids treated with adjuvants at a concentration of 0.01%, 0.10%, and 1.00% were decreased significantly by 48.17%, 51.67%, and 55.73% compared to the liquids without the adjuvants. The static contact angles of the droplets of the 120 g/hm² and 225 g/hm² acetamiprid formulations were 78.58° and 75.28°, respectively, on the adaxial leaf surface, and 89.96° and 93.74°, respectively, on the abaxial leaf surface of pittosporum tobira. The 120 g/hm² liquid foliage-applied with orange oil adjuvant significantly reduced the static contact angle of the droplets on the adaxial and abaxial leaf surfaces of pittosporum tobira. When the concentration is 1.00%, the static contact angles of the droplets of the acetamiprid liquid on the adaxial and abaxial leaf surfaces of pittosporum tobira were the lowest (19.11° and 27.19°, respectively), which were significantly decreased by 75.68% and 69.78% compared to the liquid without the adjuvants. With the concentration of the orange oil adjuvant (0.01% to 1.00%), the adhesion tension of the 120 g/hm² liquid of the acetamiprid formulations on the adaxial leaf surface of pittosporum tobira increased by 6.89–114.16%, and the adhesion work of the 120 g/hm² liquid of the acetamiprid formulations on the adaxial leaf surface of pittosporum tobira decreased by 24.26–39.07% compared to the liquid without the adjuvants. The adhesion tension on the abaxial leaf surface of pittosporum tobira increased by 226.40 to 530.00 times, and the adhesion work of on the abaxial leaf surface of pittosporum tobira decreased by 14.46–31.36% compared to the liquid without the adjuvants (see Table 2).

### 3.3. Analysis on Wetting Behaviors of Individual Pesticide Droplet on Leaves of Pittosporum Tobira

The contact angle (θ) is often used to judge the wettability of the droplet leaf surface, with θ < 60° representing good wettability, 60° ≤ θ < 80° representing moderate wettability, 80° ≤ θ < 100° representing poor wettability, and θ ≥ 100° representing very poor wettability [32]. On the leaves of pittosporum tobira, the wetting behaviors of the droplets with different physical and chemical properties are shown in Figures 2 and 3. It can be seen from the figures that the wetting behaviors of the droplets with different properties on the adaxial and abaxial leaf surfaces of pittosporum tobira were relatively similar. As time went by, the contact angle of the droplets on the leaves of pittosporum tobira decreased gradually and tended to balance within a certain time. The static contact angles ($\theta_{40}$) of the water droplets on the adaxial and abaxial leaf surfaces of pittosporum tobira were more than 80° and less than 100°, indicating poor wettability; the static contact angles ($\theta_{40}$) of the droplets of the 120 g/hm² and 225 g/hm² acetamiprid liquids were all more than 60° and less than 80°, indicating moderate wettability on the adaxial leaf surface of pittosporum tobira, and all were more than 80° and less than 100°, thus indicating poor wettability on

the abaxial leaf surface of pittosporum tobira. The static contact angles ($\theta_{40}$) of the droplets of the acetamiprid liquid with 0.01% orange oil adjuvant on the adaxial and abaxial leaf surfaces of pittosporum tobira were all more than 60° and less than 80°, indicating moderate wettability. The static contact angles ($\theta_{40}$) of the droplets of the acetamiprid liquids with 0.10% and 1.00% orange oil adjuvant on the adaxial and abaxial leaf surfaces of pittosporum tobira were all less than 40°, indicating good wettability. This shows that adding adjuvants can regulate the wetting behavior of droplets on the leaves of pittosporum tobira, and the deviation of such wetting behavior is closely related to the amounts of adjuvant.

**Table 2.** Effect of physicochemical properties of acetamiprid solution with orange oil adjuvant.

| Leaf | Dosage of Formulation (g/hm²) | Concentration of Adjuvant (%) | γ Surface Tension (mN/m) | Surface Free Energy (mJ/m²) | θ Static Contact Angle (°) | β Adhesion Tension (mN/m) | Wₐ Adhesion Work (mN/m) |
|---|---|---|---|---|---|---|---|
| Adaxial Leaf | 120 | 0.00 | 67.43 ± 0.80 b | 67.43 | 78.58 ± 2.96 a | 13.35 | 80.78 |
| | 120 | 0.01 | 34.95 ± 0.39 d | 34.95 | 65.91 ± 2.96 b | 14.27 | 49.22 |
| | 120 | 0.10 | 32.59 ± 0.98 e | 32.59 | 28.70 ± 2.98 c | 28.59 | 61.18 |
| | 120 | 1.00 | 29.85 ± 0.43 f | 29.85 | 19.11 ± 1.73 d | 28.21 | 58.06 |
| | 225 | 0.00 | 63.26 ± 0.83 c | 63.26 | 75.28 ± 11.04 a | 16.07 | 79.33 |
| | CK(Water) | - | 72.80 ± 0.40 a | 72.80 | 83.41 ± 8.14 a | 8.35 | 81.15 |
| Abaxial Leaf | 120 | 0.00 | 67.43 ± 0.80 b | 67.43 | 89.96 ± 3.78 a | 0.05 | 67.48 |
| | 120 | 0.01 | 34.95 ± 0.39 d | 34.95 | 71.01 ± 5.29 b | 11.37 | 46.32 |
| | 120 | 0.10 | 32.59 ± 0.98 e | 32.59 | 39.55 ± 3.68 c | 25.13 | 57.72 |
| | 120 | 1.00 | 29.85 ± 0.43 f | 29.85 | 27.19 ± 9.63 d | 26.55 | 56.40 |
| | 225 | 0.00 | 63.26 ± 0.83 c | 63.26 | 93.74 ± 8.66 a | −4.13 | 59.13 |
| | CK (Water) | - | 72.80 ± 0.40 a | 72.80 | 96.96 ± 7.18 a | −8.82 | 63.98 |

The data in the table are mean ± standard deviation ($n$ = 5). Different letters in the same column indicate significant difference at the level of $p < 0.05$ by Duncan's new multiple range test.

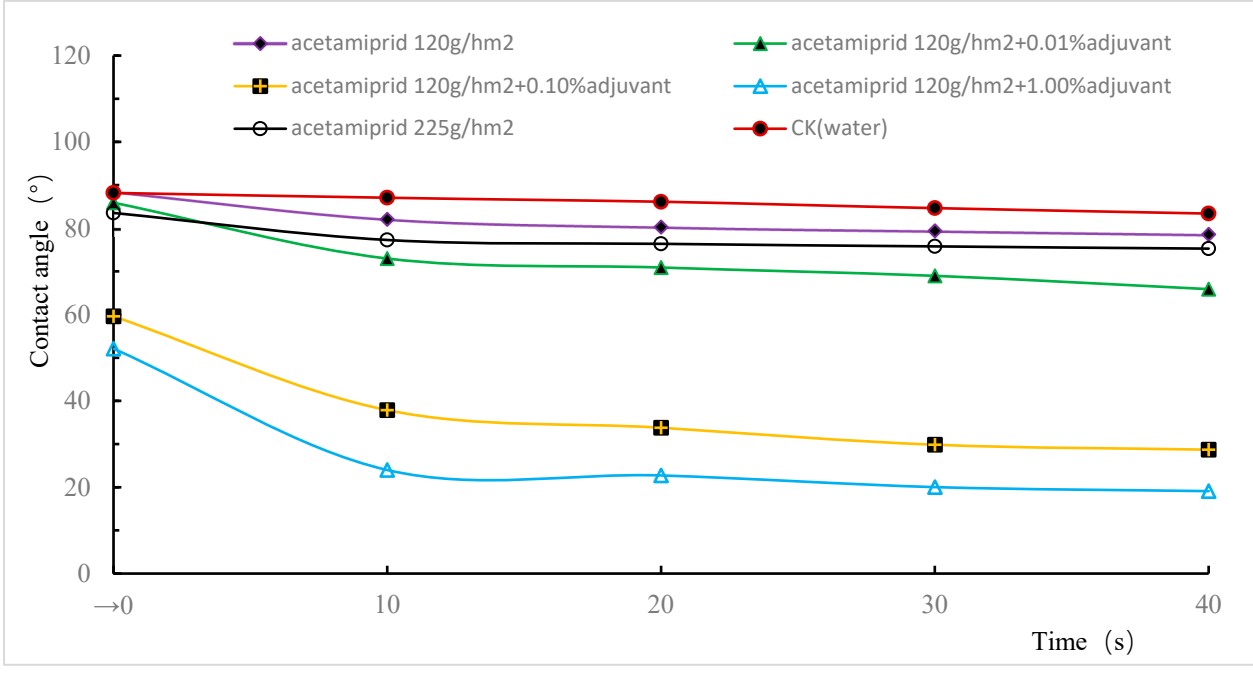

**Figure 2.** Wetting behavior of droplets of acetamiprid liquid with different amounts of adjuvants on the adaxial leaf surface of *Pittosporum tobira*.

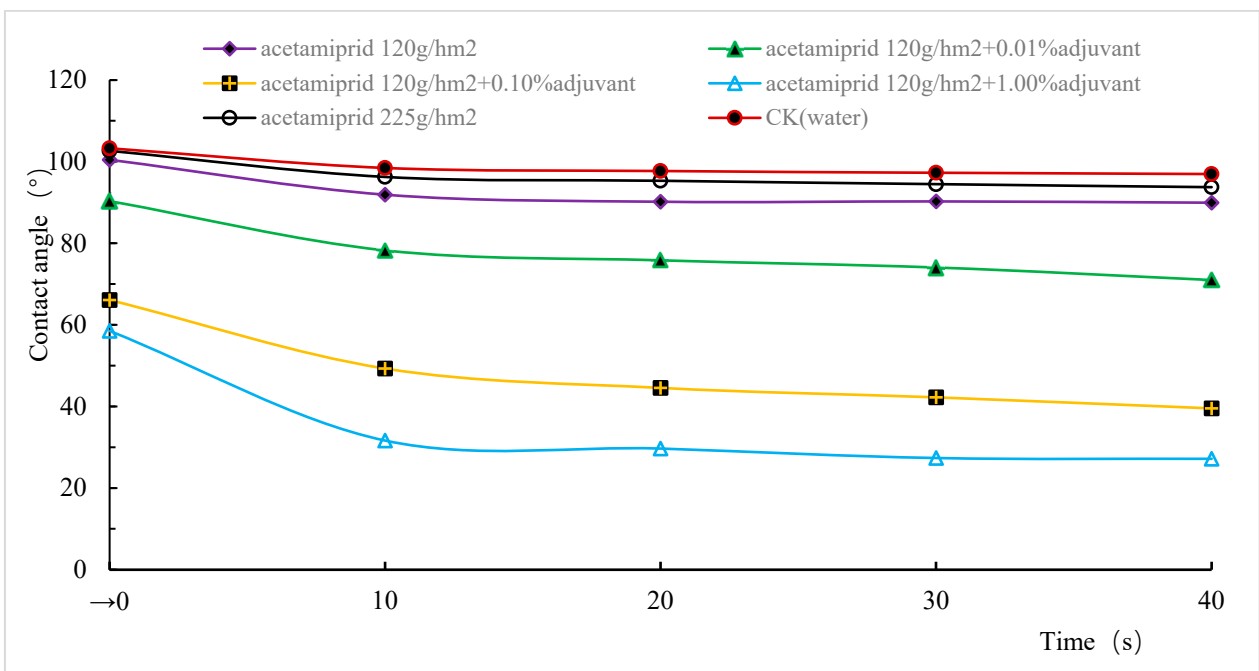

**Figure 3.** Wetting behavior of droplets of acetamiprid liquid with different amounts of adjuvants on the abaxial leaf surface of *Pittosporum tobira*.

### 3.4. Variation Rate of Contact Angle

The variation rate of the contact angle of the droplets with different properties on the adaxial and abaxial leaf surfaces of pittosporum tobira is shown in Figure 3. The calculation results on the adaxial leaf surface of pittosporum tobira showed that the variation rates of the contact angles of the 120 g/hm$^2$ acetamiprid liquids with 0.01%, 0.10%, and 1.00% adjuvant liquids were relatively higher (0.50, 0.77, and 0.83). All of them were significantly higher than those of the liquid without the adjuvants. The variation rate of the contact angle of the water droplets was 0.12, which was significantly lower than those of the 120 g/hm$^2$ and 225 g/hm$^2$ acetamiprid liquids. The calculation results on the abaxial leaf surface of pittosporum tobira showed that the variation rates of the contact angles of the 120 g/hm$^2$ acetamiprid liquids with 0.01%, 0.10%, and 1.00% adjuvant were relatively higher (0.48, 0.66, and 0.78). All of them were significantly higher than those of the liquid without the adjuvants. There was no significant difference in the variation rates of the contact angles between the 120 g/hm$^2$ and 225 g/hm$^2$ acetamiprid liquids and the water liquids. In general, the variation rate of the contact angle was affected by the amount of orange oil adjuvant and increased with the increasing amount of adjuvant (see Table 3).

**Table 3.** Effect of the variation rate of the contact angle of acetamiprid liquid with orange oil adjuvant.

| Dosage of Formulation (g/hm$^2$) | Concentration of Adjuvant (%) | Variation Rate of Contact Angle $r_{60} = (\theta^\circ_{\to 0} - \theta^\circ_{40})/40$ | |
|---|---|---|---|
| | | **Adaxial Leaf** | **Abaxial Leaf** |
| 120 | 0.00 | 0.24 ± 0.06 c | 0.26 ± 0.13 c |
| 120 | 0.01 | 0.50 ± 0.06 b | 0.48 ± 0.01 b |
| 120 | 0.10 | 0.77 ± 0.04 a | 0.64 ± 0.07 ab |
| 120 | 1.00 | 0.83 ± 0.06 a | 0.78 ± 0.11 a |
| 225 | 0.00 | 0.21 ± 0.05 c | 0.22 ± 0.18 c |
| CK (Water) | - | 0.12 ± 0.08 d | 0.16 ± 0.15 c |

The data in the table are mean ± standard deviation (*n* = 5). Different letters in the same column indicate significant difference at the level of *p* < 0.05 by Duncan's new multiple range test.

*3.5. Field Efficacy Trials*

According to the results of the field efficacy experiment, after pesticide application for 5 days, the control effect of the 225 g/hm$^2$ acetamiprid spray against *E. shanghaiensis* on pittosporum tobira was 81.63%, which was significantly higher than that of the 120 g/hm$^2$ acetamiprid spray. The control effect of the 120 g/hm$^2$ acetamiprid liquids with 0.01%, 0.10%, and 1.00% adjuvant against *E. shanghaiensis* on pittosporum tobira were significantly better than that of the 120 g/hm$^2$ acetamiprid liquid. There was no significant difference between the control effects of the acetamiprid liquids with 0.10% and 1.00% adjuvant and the 225 g/hm$^2$ acetamiprid liquid against *E. shanghaiensis* on pittosporum tobira, but they were significantly better than the 120 g/hm$^2$ acetamiprid liquid with the 0.01% adjuvant. After pesticide application for 10 days, the control effect of the 120 g/hm$^2$ acetamiprid liquid with 1.00% adjuvant against *E. shanghaiensis* on pittosporum tobira was 88.79%, which was significantly higher than that of the other treatments; there was no significant difference between the control effects of the 120 g/hm$^2$ acetamiprid liquid with 0.10% adjuvant and the 225 g/hm$^2$ acetamiprid liquid against *E. shanghaiensis* on pittosporum tobira, but they were significantly better than the 120 g/hm$^2$ acetamiprid liquid and the 120 g/hm$^2$ acetamiprid liquid with 0.01% adjuvant. The control effect of the 120 g/hm$^2$ acetamiprid liquid with 0.01% adjuvant against *E. shanghaiensis* on pittosporum tobira was better than that of the 120 g/hm$^2$ acetamiprid liquid. The analysis of the results after pesticide application for 15 days were the same as the results after 10 days, and the 120 g/hm$^2$ acetamiprid liquid with 1.00% adjuvant had the best control effect against *E. shanghaiensis* on pittosporum tobira (see Table 4).

**Table 4.** Field efficacy of acetamiprid against *E. shanghaiensis* on pittosporum tobira.

| Dosage of Formulation (g/hm$^2$) | Concentration of Adjuvant (%) | Pest Population (head) | 5 d | | 10 d | | 15 d | |
|---|---|---|---|---|---|---|---|---|
| | | | Reduction Rate of Pest Population/% | Control Effect/% | Reduction Rate of Pest Population/% | Control Effect/% | Reduction Rate of Pest Population/% | Control Effect/% |
| 120 | 0.00 | 105.0 ± 8.2 | 64.81 | 68.25 ± 0.71 c | 68.99 | 74.17 ± 2.03 d | 69.53 | 77.82 ± 1.91 d |
| 120 | 0.01 | 110.7 ± 7.1 | 69.90 | 72.85 ± 2.66 b | 74.12 | 78.47 ± 1.81 c | 76.21 | 82.66 ± 1.00 c |
| 120 | 0.10 | 107.3 ± 8.0 | 78.28 | 80.39 ± 0.63 a | 82.29 | 85.24 ± 1.47 b | 84.16 | 88.47 ± 0.93 b |
| 120 | 1.00 | 111.0 ± 12.3 | 80.81 | 82.68 ± 0.95 a | 86.53 | 88.79 ± 2.06 a | 87.61 | 90.98 ± 0.96 a |
| 225 | 0.00 | 104.7 ± 11.6 | 79.66 | 81.63 ± 1.21 a | 83.20 | 86.01 ± 0.74 b | 84.77 | 88.90 ± 0.79 b |
| CK(Water) | - | 105.7 ± 8.0 | −10.79 | - | −20.12 | - | −37.29 | - |

The data in the table are mean ± standard deviation (*n* = 3). Different letters in the same column indicate significant difference at the level of *p* < 0.05 by Duncan's new multiple range test.

**4. Discussion**

Currently, there are still disputes with regard to the concept of hydrophilic and hydrophobic interfaces. It is generally believed that the contact angle of 90° is the boundary; a solid surface with a contact angle less than 90° is classified as a hydrophilic surface, and a surface with a contact angle more than 90° is classified as a hydrophobic surface. However, some research results show that the boundary between hydrophilic and hydrophobic surfaces should be defined at about 65° [33,34], and the results analyzed according to that boundary have expanded the range of hydrophobic surfaces compared with the analyses in which 90° is defined as the boundary. In this study, the static contact angles of water on the adaxial and abaxial leaf surfaces of pittosporum tobira were all more than 65°, indicating that it is a hydrophobic interface, which is different from the original definition of being hydrophilic on the adaxial leaf and hydrophobic on the abaxial leaf of pittosporum tobira. Based on this boundary, the leaf surface of most varieties of maize belongs to the hydrophilic surface, except for the abaxial leaf surface of the NW Spanish maize population, with a water contact angle of 75.35° [35]. For a hydrophobic interface, it is often necessary to spray liquid with adjuvants to adjust the physical and chemical properties of the pesticide solution, so as to improve the efficacy when using the pesticide spray with water to control pests [36]. According to Young's equation, only when the SFE of the liquid is less than that of the leaf surface can it be fully wetted [37]. The SFE value of the leaf surfaces of pittosporum tobira is about 38.95 mJ/m$^2$, and the SFE value should

be less than 38.95 mJ/m$^2$. The acetamiprid liquid should be foliage-applied with spray adjuvants because of the higher SFE value. The ability and efficiency of the adjuvants to reduce surface tension should be considered in foliar spray processes [38]. Some studies have shown that the apple tree leaf surface reached a completely wet state, and the control efficacy of beta-cyfluthrin against *Carposina niponensis* was significantly improved with the concentration of nonionic surfactants C$_{12}$E$_5$ and Triton X-100 above CMC [39]. For some surfactants, the concentration above CMC is necessary for complete retention on the *Hordeum vulgare* leaf surface or a decrease in the height of the bouncing drops on the rice leaf surface [40,41]. Our results also confirm this conclusion. In this study, the surface tension of acetamiprid liquids with the concentrations of adjuvant (0.10% and 1.00%) above the CMC decreased by 51.67–55.73%, resulting in the surface energy (equal to the surface tension value at room temperature and under atmospheric pressure) to be lower than the surface free energy of the leaf surface of pittosporum tobira. Moreover, the adhesion tension and wettability of the droplets on the leaf surface of pittosporum tobira also improved. Therefore, the control effect of the acetamiprid liquid with the same dosage against *E. shanghaiensis* on pittosporum tobira can be significantly improved with orange oil adjuvant, and it is equivalent or even better than that of high doses of acetamiprid liquid against *E. shanghaiensis* on pittosporum tobira, which provides the basis for the scientific pesticide control against *E. shanghaiensis* on pittosporum tobira.

*E. shanghaiensis* is a pest newly found to harm pittosporum tobira and few reports have studied and evaluated the control of the pest. In this paper, the optimal time for spraying is when the damage caused by the nymphs has not yet led to leaf curling and longitudinal wrapping. During the process of chemical controls, once the leaves of pittosporum tobira become curled longitudinally due to pest damage and are directly wrapped around the nymphs of *E. shanghaiensis*, the probability of sprayed droplets coming into contact with the target pests is reduced, which is not conducive to the effect of contact-type pesticides. Acetamiprid is a nicotinamide insecticide synthesized based on the nitromethylene compound, featuring contact kill, stomach poisoning, powerful systemic absorption, and a certain degree of permeation, which is considered to be a good pesticide for psyllid control [42,43]. The field efficacy experiment showed that the control effect of 120 g/hm$^2$ acetamiprid against *E. shanghaiensis* on pittosporum tobira was 77.82% after pesticide application for 15 days and increased by 6.22–16.91% with orange oil adjuvants. The addition of orange oil adjuvants improved the adhesion force and wettability of the pesticide liquid on the leaf surface of pittosporum tobira to a certain extent, and thus achieved a better control effect. As highly active substances, spray adjuvants are commonly used in low amounts to adjust the physical and chemical properties of pesticide liquids. In addition, the reasonable use of spray adjuvants can achieve the goals of pesticide amount reduction and synergistic effect. Therefore, the scientific use of spray adjuvants is an essential way to reduce the amount of pesticides that need to be used [44–49].

In the deposition process of pesticide target spraying, it is very important to select the matching pesticide adjuvants and the corresponding addition amounts according to the biological characteristics of the target plants. Insufficient contents of wettability adjuvants in pesticide formulations, or improper types of adjuvants selected during the process of the spray control of hydrophobic plants, can lead to a serious loss of pesticide liquid on the target during the spraying process, which can reduce the utilization of the pesticides. To this end, a surface free energy database relevant to target crops is established based on the biological characteristics of the leaves of target plants under specific scenarios of pesticide application so as to guide the scientific use of adjuvants in production. This will be conducive to the effective deposition of pesticides in target spraying, improve the utilization of pesticides, and thus reduce the amount of pesticide that needs to be used.

**Author Contributions:** Conceptualization, G.X. and A.C.; methodology, G.X.; software, D.Y.; validation, D.X. and L.X.; formal analysis, W.F.; investigation, G.X.; resources, Q.W.; data curation, L.X.; writing—original draft preparation, G.X.; writing—review and editing, D.Y.; visualization, D.X.; supervision, Q.W.; funding acquisition, A.C. All authors have read and agreed to the published version of the manuscript.

**Funding:** This research was funded by the National Key Research and Development Program of China, grant number 2017YFD0201600.

**Institutional Review Board Statement:** Not applicable.

**Informed Consent Statement:** Not applicable.

**Data Availability Statement:** Not applicable.

**Conflicts of Interest:** The authors declare no conflict of interest.

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
