# Peer review of "Synergistic Effect of Orange Oil Adjuvant on Acetamiprid in the Control of Edentatipsylla shanghaiensis"

_sustainability, doi:10.3390/su151310113_

Round 1

Reviewer 1 Report

This manuscript investigated the effect of orange oil adjuvant on the physicochemical properties of pesticide solutions and the control efficacy on Edentatipsylla shanghaiensis. The authors found that the insecticidal effect of acetamiprid against E. shanghaiensis was significantly improved after spraying with orange oil adjuvant. The results are useful and proved a new way for farmers to control for E. shanghaiensis. The article has clear ideas and data analysis in place. Before published, some small questions should be addressed.

Abstract

1. Line 14, deleted” Field efficacy tests were also carried out.

2. Line 23, at the end of the abstract, adding the significance your finding, such as “Our results proved a new way for farmers to control for E. shanghaiensis.

Introduction

3. Line 53, is Figure 1 taken by the author himself? If no, please provide its source.

Methods and materials

4. Line 97-102, “0-50μL” revised to “0-50 μL”, “0.3MPa” revised to “0.3 MPa”, “0.7L/min” revised to “0.7 L/min”, there is a space between the number and the unit. Please check the full text.

Results

5. Line 219, P should be lower-case and italic. Please check and revise the full text.

Table

6. Line 217,268,272, please check the CK tremeatment in the table2, table3 and table4 is consistent. If the same is the same statement.

This manuscript investigated the effect of orange oil adjuvant on the physicochemical properties of pesticide solutions and the control efficacy on Edentatipsylla shanghaiensis. The authors found that the insecticidal effect of acetamiprid against E. shanghaiensis was significantly improved after spraying with orange oil adjuvant. The results are useful and proved a new way for farmers to control for E. shanghaiensis. The article has clear ideas and data analysis in place. Before published, some small questions should be addressed.

Abstract

1. Line 14, deleted” Field efficacy tests were also carried out.

2. Line 23, at the end of the abstract, adding the significance your finding, such as “Our results proved a new way for farmers to control for E. shanghaiensis.

Introduction

3. Line 53, is Figure 1 taken by the author himself? If no, please provide its source.

Methods and materials

4. Line 97-102, “0-50μL” revised to “0-50 μL”, “0.3MPa” revised to “0.3 MPa”, “0.7L/min” revised to “0.7 L/min”, there is a space between the number and the unit. Please check the full text.

Results

5. Line 219, P should be lower-case and italic. Please check and revise the full text.

Table

6. Line 217,268,272, please check the CK tremeatment in the table2, table3 and table4 is consistent. If the same is the same statement.

Author Response

We thank reviewer 1 for the constructive comments on our manuscript. We really appreciate your efforts in reviewing our manuscript and have revised the manuscript accordingly. Our point-by-point responses are detailed in below.

Abstract

  1. Line 14, deleted”Field efficacy tests were also carried out.”

Response: Done. The advice of other reviewers was adopted and we have rewritten the abstract (in Page 1, Line 9-26 in the revised manuscript).

  1. Line 23, at the end of the abstract, adding the significance your finding, such as “Our results proved a new way for farmers to control for shanghaiensis.

Response: Done. We have rewritten the abstract and expressed the importance of our finding (in Page 1, Line 9-26 in the revised manuscript).

Introduction

  1. Line 53, is Figure 1 taken by the author himself? If no, please provide its source.

Response: Yes, we took Figure 1 ourselves. And we have provided its source in Page 1, Line 44-45 in the revised manuscript.

Methods and materials

  1. Line 97-102, “0-50μL” revised to “0-50 μL”, “0.3MPa” revised to “0.3 MPa”, “0.7L/min” revised to “0.7 L/min”, there is a space between the number and the unit. Please check the full text.

Response: Done, we checked the full text and confirmed that there is a space between the number and the unit in the revised manuscript.

Results

  1. Line 219, P should be lower-case and italic. Please check and revise the full text.

Response: Done, we made sure that the P’s were all lowercase and italic in full text in the revised manuscript.

Table

  1. Line 217,268,272, please check the CK tremeatment in the table2, table3 and table4 is consistent. If the same is the same statement.

Response: Done. We checked the CK tremeatment and revised CK to CK (Water) in the table2, table3 and table4 in the revised manuscript.

Reviewer 2 Report

 Dear,

The manuscript entitled ʻSynergistic Effect of Orange Oil Adjuvant on Acetamiprid in the Control of Edentatipsylla shanghaiensisʼ has interesting subject and useful findings about reducing the use of chemical pesticides. However, some modifications are necessary as follows:

Line 10: Add the common name and full scientific name of insect pest at first mention.

Line 15: Combine this sentence with the previous one.

Line 23: Write the concluding sentence at the end of the abstract.

Line 25: Remove ʻLi et Chenʼ.

Line 29: Remove ʻIt is a serious pest on pittosporum tobiraʼ according to previous and next sentences.

Line 33: Remove ʻLi et Chen ʼ and write the abbreviation of scientific name after first mention and in the middle of sentence; E. shanghaiensis. Consider this throughout the manuscript (see for example lines 40, 46 and 85).

Line 35: Where? Add a reference.

Figure 1: Write the information about the figure. Where, on what date and on what plant was the photo taken?

Line 84: Add suitable background about the possibility of using plant oils as adjuvants. It is the main objective of this research.

Line 90: Remove ʻin this experimentʼ.

Line 110: Add related references for this sentence and the method.

Line 140: Remove ʻfor “Aisheng” (a variety of pittosporum tobira)ʼ. These words are written a few lines above and repeated.

Line 145: Rewrite the sentences ʻAdd … water blank controlʼ. Sentences are written in imperative form. This problem can be seen in some other parts of the manuscript, which should be corrected (see also the data anaysis).

Line 146: Geographic coordinates are required.

Table 1: Compare the means for each row.

Line 328-330: This sentence is completely repetitive and is stated in the introduction section.

 Best regard

Extensive editing of English language required. Please see the above comments. 

Author Response

We really appreciate your recognition and suggestions for our manuscript. We believe your comments will benefit us in our future writing. We made a substantial revision in the revised manuscript according to your comments. Our responses to your comments are as below.

1、Line 10: Add the common name and full scientific name of insect pest at first mention.

Response: We have added the common name and full scientific name of insect pest at first mention in Page 1, Line 14 in the revised manuscript.

2、Line 15: Combine this sentence with the previous one.

Response: Thanks. The advice of other reviewers was adopted and we have rewritten the abstract (in Page 1, Line 9-26 in the revised manuscript).

3、Line 23: Write the concluding sentence at the end of the abstract.

Response: We have rewritten the abstract and expressed the concluding sentence at the end of the abstract (in Page 1, Line 9-26 in the revised manuscript).

4、Line 25: Remove ʻLi et Chenʼ.

Response: Revised.

5、Line 29: Remove ʻIt is a serious pest on pittosporum tobiraʼ according to previous and next sentences.

Response: Revised.

6、Line 33: Remove ʻLi et Chen ʼ and write the abbreviation of scientific name after first mention and in the middle of sentence; E. shanghaiensis. Consider this throughout the manuscript (see for example lines 40, 46 and 85).

Response: We have written the abbreviation of scientific name after first mention and in the middle of sentence, E. shanghaiensis. The full text has been revised and checked.

7、Line 35: Where? Add a reference.

Response: E. shanghaiensis. is a new pest discovered in recent years, so we can hardly find relevant literature. The conclusion here is the result of our investigation in Jiangsu Province in the past two years.

8、Figure 1: Write the information about the figure. Where, on what date and on what plant was the photo taken?

Response: We have provided its source include date and plant in Page 1, Line 44-45 in the revised manuscript.

9、Line 84: Add suitable background about the possibility of using plant oils as adjuvants. It is the main objective of this research.

Response: Thanks. The main objective of this research is about the possibility of using plant oils as adjuvant. To illustrate our point, we added a suitable background about the plant oils as spray adjuvant in Page 3, Line86-96 in the revised manuscript.

Recently, the essential oil adjuvants are used in the agricultural field because of their safety and excellent surface activity [22-23]. The orange oil adjuvant can be used with insecticides, fungicides, and herbicides against pests [24-25]. But limited studies have reported that use the higher or lower rates of the adjuvant on different leaf surfaces to wet and how to combine with insecticides for special pest control. The relationship between the physicochemical properties of pesticide liquid and leaf surface characteristics, the wettability and field efficacy were detemined to elucidate the internal mechanism of its synergistic effect. Together with the synergistic interactions between orange oil adjuvant and widely used insecticide, the results could provide insights that suitable amounts of adjuvant used in the chemical control of E. shanghaiensis can reduce pesticide dosage and increase efficiency.

10、Line 90: Remove ʻin this experimentʼ.

Response: Revised.

11、Line 110: Add related references for this sentence and the method.

Response: Related references as below for this sentence and the method were added in Page 3, Line 121 in the revised manuscript.

26、Jañczuk, B; Bialopiotrowicz, T; Zdziennicka, A. Some remarks on the components of the liquid surface free energy. Journal of Colloid and Interface Science, 1999, 211: 96-113.

27、Fernandez, V.; Khayet, M. Evaluation of the surface free energy of plant surfaces: toward standardizing the procedure. Frontier in Plant Science, 2015, 6: 510. DOI:10.3389/fpls.2015.00510

12、Line 140: Remove ʻfor “Aisheng” (a variety of pittosporum tobira)ʼ. These words are written a few lines above and repeated.

Response: Revised.

13、Line 145: Rewrite the sentences ʻAdd … water blank controlʼ. Sentences are written in imperative form. This problem can be seen in some other parts of the manuscript, which should be corrected (see also the data anaysis).

Response: Done. This sentences revised to “The 120 g/hm2 spray liquid containing 50% acetamiprid water dispersible granules was foliage-applied with orange oil adjuvant at the concentration of 0.01%, 0.10%, and 1.00%.” The same problem the other parts of the manuscript also revised.

14、Line 146: Geographic coordinates are required.

Response: Geographic coordinates are added in Page 4, Line 153-154 in the revised manuscript.

15、Table 1: Compare the means for each row.

Response: We compared the means for each row and the description of statistical analysis was in Page 5, Line 191-194 in the revised manuscript.

16、Line 328-330: This sentence is completely repetitive and is stated in the introduction section.

Response: According to the suggestions, this sentence is completely repetitive and is stated in the introduction section, so we deleted it.

Reviewer 3 Report

The subject of the paper is interesting. The manuscript is well planned however, the Authors should pay attention to some aspects listed below:

- In the introduction section, Authors should clearly highlight the novelty of their work

- The analysis should be supplemented with a comparison of the obtained results with literature references and results of other researchers

- In the abstract section, the Authors should clearly express the importance of this manuscript

- Many abbreviations are used in the manuscript, authors should make sure all are explained

- Equation 5 and 6 should be extracted from the text and centered

- Figures in color would be more readable

Author Response

We thank Reviewer for the constructive comments on our manuscript. We really appreciate your efforts in reviewing our manuscript and have revised the manuscript accordingly. Our point-by-point responses are detailed in below.

Responds to the reviewers’ comments:

1、- In the introduction section, Authors should clearly highlight the novelty of their work

Response: We added a suitable background about the plant oils as spray adjuvant and clearly highlight the novelty of our work in Page 3, Line 86-96 in the revised manuscript. At the same time, related literature has been added.

2、- The analysis should be supplemented with a comparison of the obtained results with literature references and results of other researchers

Response: We have been supplemented with a comparison of the obtained results with literature references and results of other researchers in Page 9, Line 327-335 in the revised manuscript. At the same time, related literature has been added.

3、- In the abstract section, the Authors should clearly express the importance of this manuscript

Response: Thanks. We have rewritten the abstract and expressed the concluding sentence at the end of the abstract (in Page 1, Line 9-26 in the revised manuscript).

4、- Many abbreviations are used in the manuscript, authors should make sure all are explained

Response: We confirmed the abbreviations used in the manuscript.

5、- Equation 5 and 6 should be extracted from the text and centered

Response: Revised.

6、- Figures in color would be more readable

Response: We have reproduced Figure 2 and 3 in a color and enhanced version in the revised manuscript.

Reviewer 4 Report

Way to many compound and run on sentences.  Advise the authors to hire and/or have a bunch of English speaking individuals offer comment.

Author Response

Way to many compound and run on sentences. Advise the authors to hire and/or have a bunch of English speaking individuals offer comment.

Response: We thank the reviewer 4 for pointing this out. We once again sought the help of professional language editing services in related fields and corrected our English language in the revision.

The research on the relationship between the interface behavior of pesticide droplet and its biological activity has always been a hot topic for scientists. The bounce of the pesticide droplets reduces their contact probability to the target pests and thus reduces their effectiveness. The effect of pesticide is directly related to the leaf wetting and adhesion ability of the liquid on the target plants, and the adhesion ability is related to the surface free energy of the liquid and the contact Angle of the liquid on the leaf surface. The scientific application of pesticide adjuvants can effectively regulate the foliar behavior of pesticide droplets. However, the properties of the adjuvant, the ability to reduce surface free energy and the properties of the target must be taken into account. Our research aims to get suitable amounts of adjuvant used in the chemical control of E. shanghaiensis according to leaf wetting and the adhesion ability of the liquid can reduce pesticide dosage and increase efficiency.

We really appreciate for your approval of our manuscript. Nevertheless, there are actually some shortcomings in our manuscript, and we made a substantial revision according to the professional comments in this revised manuscript. We hope that the revised manuscript will meet the requirements for publication.

Revise abstract:

We explore the effect of orange oil adjuvant (a kind of spray adjuvants) on the physicochemical properties of acetamiprid (pesticide) when foliage-applied to the surface of pittosporum tobira leaves. The leaf surface was characterized by the OCG (Van Oss-Chaudhury-Good) method and the relationship between wetting behavior of pesticide droplets including the change in surface free energy (SFE), adhesion force, and adhesion work is explored to offer insight into the control of the pittosporum tobira psyllid, Edentatipsylla shanghaiensis Li et Chen. Results showed that SFE values for the adaxial and adaxial leaf surfaces were 40.13 mJ/m² and 37.06 mJ/m², respectively, while acetamiprid liquids had SFE values of 67.43 mJ/m² and 63.26 mJ/m². SFE values of the acetamiprid liquids are greater than that of the leaf surface, and the droplets on the leaves with smaller adhesion force and lager adhesion work exhibited moderate to poor wettability estimated by contact angles. When the concentration of the orange oil adjuvant was between 0.10% and 1.00% above CMC (critical micellar concentration, 0.09%), the SFE values of the acetamiprid liquids were less than that of the leaf surface. The adhesion force was greatly increased, and the adhesion work decreased by 14.46%-28.13%. Meanwhile, droplets on the leaves exhibited good wettability. Field experiments showed that the insecticidal effect of acetamiprid against E. shanghaiensis was significantly improved after spraying with orange oil adjuvant at the concentration (0.10%, 1.0%) above CMC. The results can provide reference for reducing pesticide dosage and increasing efficiency during the chemical control of pests.

Round 2

Reviewer 2 Report

Dear,

All requested amendments have been considered. The modifications made are acceptable and I have no new suggestions.

Best regards

Author Response

We would like to take this opportunity to express my sincere gratitude for your time and efforts in reviewing my manuscript. Your positive feedback and encouragement have been instrument in enhancing the quality of final manuscript, and I am truly honored to have had the benefit of your expertise and insights. Thank you!